# Alterations of the Gut Microbiota and Metabolomics Associated with the Different Growth Performances of *Macrobrachium rosenbergii* Families

**DOI:** 10.3390/ani13091539

**Published:** 2023-05-04

**Authors:** Xuan Lan, Xin Peng, Tingting Du, Zhenglong Xia, Quanxin Gao, Qiongying Tang, Shaokui Yi, Guoliang Yang

**Affiliations:** 1Zhejiang Provincial Key Laboratory of Aquatic Resources Conservation and Development, Key Laboratory of Aquatic Animal Genetic Breeding and Nutrition, Chinese Academy of Fishery Sciences, College of Life Sciences, Huzhou University, Huzhou 313000, China; lanxuan0570@163.com (X.L.); pengxin0521@163.com (X.P.); t_sep24@163.com (T.D.); gaoqx2008@163.com (Q.G.); tangqy@zjhu.edu.cn (Q.T.); 2Jiangsu Shufeng Prawn Breeding Co., Ltd., Gaoyou 225654, China; zjhill@126.com

**Keywords:** growth performance, gut bacteria, metabolism, LC–MS

## Abstract

**Simple Summary:**

This study used 16S rRNA sequencing and metabolomic methods to investigate the key gut microbiota and metabolites associated with the growth performance of *Macrobrachium rosenbergii* families. We found that some critical intestinal bacteria, including *Lactobacillus* and *Blautia*, and the metabolites related to metabolism of amino acids and fatty acids may play critical roles in the growth of prawns. This study contributes to figuring out the landscape of the gut microflora and intestinal metabolites associated with prawn growth performance and provides a basis for future studies on the probiotic feed of prawns.

**Abstract:**

To investigate the key gut microbiota and metabolites associated with the growth performance of *Macrobrachium rosenbergii* families, 16S rRNA sequencing and LC–MS metabolomic methods were used. In this study, 90 *M. rosenbergii* families were bred to evaluate growth performance. After 92 days of culture, high (H), medium (M), and low (L) experimental groups representing three levels of growth performance, respectively, were collected according to the weight gain and specific growth rate of families. The composition of gut microbiota showed that the relative abundance of Firmicutes, Lachnospiraceae, *Lactobacillus,* and *Blautia* were much higher in Group H than those in M and L groups. Meanwhile, compared to the M and L groups, Group H had significantly higher levels of spermidine, adenosine, and creatinine, and lower levels of L-citrulline. Correlation analysis showed that the abundances of *Lactobacillus* and *Blautia* were positively correlated with the levels of alpha-ketoglutaric acid and L-arginine. The abundance of *Blautia* was also positively correlated with the levels of adenosine, taurine, and spermidine. Notably, lots of metabolites related to the metabolism and biosynthesis of arginine, taurine, hypotaurine, and fatty acid were upregulated in Group H. This study contributes to figuring out the landscape of the gut microbiota and metabolites associated with prawn growth performance and provides a basis for selective breeding.

## 1. Introduction

Among the most important crustacean species in aquaculture throughout tropical and subtropical regions, the giant freshwater prawn (GFP; *Macrobrachium rosenbergii*) is one of the world’s most important economic freshwater crustaceans; it has rapid growth rates, a wide diet, a short breeding cycle, rich meat nutrition, and good adaptability [1]. Furthermore, it is the largest freshwater prawn in the world [2]. The cultivation of *M. rosenbergii* has been increasingly important in enhancing agricultural efficiency and farmer incomes in China. In 2021, the production of *M. rosenbergii* reached 171,000 tons, and it is widely farmed in parts of southern China such as Jiangsu, Zhejiang, Guangdong, etc. However, with the growing demand for *M. rosenbergii*, the supply of genetically improved varieties is causing a bottleneck in *M. rosenbergii* aquaculture. Hence, the breeding of new varieties with improved traits, including stress tolerance, high growth rates, and low feed coefficient, is extremely urgent for the industry.

The animal intestine is a complex ecosystem [3] and the primary organ of digestion and absorption [4]. Many bacteria inhabit aquatic animals’ intestinal tracts, and the gut microbiota plays a critical role in determining the phenotypes of aquatic animals, including nutrient absorption [5], metabolism [6], energy consumption [7], development [8], immunity [9], disease resistance [3,10], etc. Previous studies showed that intestinal bacteria composition is also affected by host factors such as dietary conditions, development level, and growth performance [11,12]. A relatively stable level of diversity can be maintained in gut bacteria by the host’s gut immune system and its intestinal environment [13]. For example, Xiao et al. [14] discovered that the intestinal flora of zebrafish (*Danio rerio*) was separated in different developmental stages. Zhang et al. [15] explored the gut microbial community of *Procambarus clarkii*, and revealed that gut microbiota patterns of crayfish were altered by diet and development. Wang et al. [16] explored the relationship between the gut microbiota and body mass and suggested that the ratio of Bacteroidetes to Firmicutes and alpha diversity indices of gut microbiota might be related to the body mass of grass carp (*Ctenopharyngodon idella*). Xiong et al. [17] conducted 16S rRNA sequencing on gut and seawater samples of healthy and diseased *Litopenaeus vannamei*, and the results showed that changes in the intestinal bacterial community were closely related to the severity of diseased shrimp. Duan et al. [18] investigated the composition of the intestinal microbiome of *L. vannamei* fed with diets containing *Clostridium butyricum*, and found that the microbial metabolism activity was enhanced, and the digestion and immunity of the host were promoted.

Over the past decades, many studies have demonstrated that the structure of gut microbiota could affect the growth and development of the host organisms [19,20]. For example, Fan et al. [19] found that different gut microbiota affected nutrient absorption and body weight of *L. vannamei*. As a result of regulating feed conversion efficiency, intestinal flora could promote intestinal metabolism, which is of great importance for cultivating new strains with excellent growth properties [20]. Sha et al. [21] characterized the microbial community of the intestine in *Apostichopus japonicus* and showed that the growth performance of sea cucumbers was closely related to the gut microbiota. How gut microbiota affect feed conversion efficiency, and whether they contribute to growth and development, nutrition metabolism, immunity, and disease resistance still need to be further explored.

Intestinal metabolites in aquatic animals have been extensively investigated in recent years. Metabolomic methods have been used to efficiently acquire and analyze large quantities of intestinal metabolites [22]. Uengwetwanit et al. [23] integrated 16S rRNA sequencing and high-performance liquid chromatography–mass spectrometry (LC–MS) to investigate the gut microbiota and metabolites involved in the growth performance of black tiger shrimp (*Penaeus monodon*) by comparing the top- and the bottom-weight shrimp. Likewise, Guo et al. [24] found that the addition of potentially beneficial bacteria would promote the accumulation of various bioactive metabolites and benefit the growth of shrimp. Chen et al. [25] showed that the reduced growth rate and increased intestinal permeability of Chinese seabass (*Lateolabrax maculatus*) were related to the changes in intestinal bacterial microbiota and metabolites. A previous study proposed that changes in gut microbiota of fish could alter the metabolism of tryptophan, which in turn affects intestinal physiological functions [26]. In marine medaka (*Oryzias melastigma*), He et al. [27] found that some genera (such as *Shewanella* and *Haloferula*) have the capabilities of converting energy, bioremediating, and detoxifying, which has important effects on the metabolic health of marine medaka.

Gut microbiota indeed plays a crucial role in intestinal development, host growth, and immunity of aquatic animals. In addition, the gut microbiota interacts with the host primarily through the produced metabolites. The dynamics of gut microbiota can affect the intestinal metabolic network as well. The roles of intestinal microbiota and metabolites in high-performance GFP families are still not clear. To investigate the features of gut microbiota and metabolites in the GFP families with different growth performance, we used 16S rRNA sequencing and LC–MS methods to explore the intestinal bacteria composition and abundance, as well as the characteristics of intestinal metabolites of 90 GFP families. This study contributes to understanding the superiority of gut microbiota and metabolites of GFP families with high growth rates, and provides a better understanding of the improvement of the digestion and absorption capacity of GFP in aquaculture.

## 2. Materials and Methods

### 2.1. Cultivation Management

The GFP samples were collected from Jiangsu Shufeng Aquatic Seed Industry Co., Ltd. (Gaoyou, China). A total of 90 bred families were raised, and 150 juveniles (initial body weight of 2.39 ± 0.29 g) of each family were selected for this study. Since July 2021, 90 families were cultured separately in 90 cement tanks, and the area of each cement tank was 15 m^2^. During the feeding period, the water temperature was kept at 28 ± 2 °C. The water quality was monitored regularly every day to keep the pH at 7.5–8, dissolved oxygen ≥ 5.5 mg/L, ammonia nitrogen ≤ 0.25 mg/L, and nitrite ≤ 0.1 mg /L. Replacement of fresh water was carried out every four days. The whole tank was oxygenated for 24 h without interruption. The prawns were fed with a formula diet from Jiangsu Fuyuda Grain Products Co., Ltd. (the daily feeding quantity is approximately 6–8% of the total body weight of the juveniles per tank) twice daily (06:30 and 16:00). The main ingredients of the formula diet were as follows: crude protein ≥ 38%, crude fat ≥ 5%, crude fiber ≤ 7%, ash ≤ 16%, water ≤ 12%. Particularly, from 1 September to the end of culture, formula diet was fed at 06:30 and snail meat was fed at 16:00. During the breeding period, formula diet was fed from 1 July to 31 August. From 1 September to the end of breeding, formula diet was fed at 06:30 and snail (*Bellamya quadrata*) meat was fed at 16:00. The feeding experiment lasted 92 days. After the feeding experiment, all individuals were measured for evaluating growth performance, including body weight and body length.

### 2.2. Experimental Groups

A total of 90 GFP breeding families were divided into high, medium, and low growth performance groups (named as H, M, and L) according to the weight gain and specific growth rate. Three families of each group were randomly selected for the 16S rRNA sequencing and metabolomics.

### 2.3. Sampling

Before the sampling, prawns from the 9 selected GFP families were starved for 24 h. The intestines of 28 individuals from each family were randomly sampled in a sterile environment. Subsequently, the intestinal samples were separated into 2 tubes and frozen in liquid nitrogen. Of these, 0.3 g was placed in a 5 mL sterile tube and immediately stored at −80 °C for 16S rRNA sequencing. In addition, 0.2 g was placed in a 10 mL sterile tube and stored at −80 °C for metabolomic study. For 16S rRNA sequencing and metabolomics, 6 biological replicates were performed in this study.

### 2.4. 16S rRNA Sequencing

#### 2.4.1. DNA Extraction, 16S rRNA Amplification, and Sequencing

The genome DNA of the intestines was extracted using the CTAB method. DNA concentration and purity were checked by Qubit and 1% agarose gel electrophoresis. PCR amplification was carried out using the specific primers 338F and 806R for the V3-V4 region of the bacterial 16S rRNA gene (468 bp) (338F: 5′-ACTCCTACGGGAGGCAGCAG-3′; 806R: 5′-GGACTACHVGGGTWTCTAAT-3′). All PCR reactions were carried out with 15 µL of Phusion^®^ High-Fidelity PCR Master Mix (New England Biolabs). PCR products were purified and pooled using the Qiagen Gel Extraction Kit (Qiagen, Hilden, Germany). Sequencing libraries were prepared using the TruSeq^®^ DNA PCR-Free Sample Preparation Kit (Illumina, San Diego, CA, USA) following the manufacturer’s instructions. The library quality was assessed on the Qubit 2.0 Fluorometer (Thermo Scientific, Waltham, MA, USA) and Agilent Bioanalyzer 2100 system (Thermo Scientific, USA). The library was sequenced on an Illumina NovaSeq 6000 platform with PE 250 mode at Beijing Nova Technology Co., Ltd. (Beijing, China).

#### 2.4.2. Bioinformatics and Statistical Analysis of 16S rRNA Sequencing

Bioinformatics analysis of 16S rRNA sequencing was performed with QIIME v.1.9.1 [28]. Firstly, the raw data were subjected to merging using FLASH v1.2.7 [29], and then we filtered and removed the chimera sequence to obtain the Effective Tags. All Effective Tags were clustered into operational taxonomic units (OTUs, 97% similarity level) using the Uparse v7.0.1001 [30]. The most abundant sequences were selected as the representative sequences of OTUs. In addition, to explore the differences in community composition among different groups, OTUs abundance, calculation of alpha diversity and beta diversity, Venn diagram, LEfSe (LDA effect size), and metastat analysis were performed. Observed species, Chao1, Shannon, Simpson, ACE, and PD whole tree indexes were calculated using QIIME software. The differences in the alpha diversity index between groups were analyzed using R [31] and visualized as box plots. Principal component analysis (PCA) and principal coordinates analysis (PCoA) were displayed using the “ade4” and “ggplot2” packages in R. Nonmetric multidimensional scaling (NMDS) analysis was performed using the “vegan” package in R. In addition, LEfSe analysis was performed with the LEfSe software [32] to detect differences between groups.

### 2.5. Metabolomic Analyses

#### 2.5.1. Identification of Metabolites

A total of 25 mg tissues of each group were extracted by directly adding 800 µL of precooled extraction reagent (methanol: acetonitrile: water (2:2:1, *v*/*v*/*v*)), and an internal standards mix was added for quality control. After homogenizing for 5 min using TissueLyser (JXFSTPRP, Shanghai, China), samples were then sonicated for 10 min and incubated at −20 °C for one hour. Samples were centrifuged for 15 min at 25,000 rpm, and the supernatant was then transferred for vacuum freeze drying. After centrifuging for 15 min at 25,000 rpm, the metabolites were resuspended in 600 µL of 10% methanol and sonicated for 10 min at 4 °C. The supernatants were transferred to autosampler vials for LC–MS analysis. A quality control (QC) sample was prepared by pooling the same volume of each sample to evaluate the reproducibility of LC–MS analysis. To obtain more reliable results, the samples were randomly ordered to reduce system errors. A QC sample was interspersed for every 10 samples.

#### 2.5.2. LC–MS Data Processing and Identification of Differential Metabolites

Metabolites were separated and detected using a Waters 2D UPLC (Waters, Milford, MA, USA) tandem Q Exactive high-resolution mass spectrometer (Thermo Fisher Scientific, Waltham, MA, USA). The MS raw data (raw file) collected by LC–MS were imported into Compound Discoverer 3.1 (Thermo Fisher Scientific, USA) for data processing, including peak extraction, retention time correction within and between groups, additive ion pooling, missing value filling, background peak labeling, and metabolite identification. Metabolites were identified using the BMDB (Bovine Metabolome Database), mzCloud, and ChemSpider (HMDB, KEGG, LipidMaps) databases. The main parameters of metabolite identification included precursor mass tolerance < 5 ppm, fragment mass tolerance < 10 ppm, and RT tolerance < 0.2 min.

The metaX software [33] was used for the subsequent analyses, including normalizing the data using the probabilistic quotient normalization (PQN) to obtain the relative peak area, correcting the batch effect using QC-RLSC (quality-control-based robust LOESS signal correction), calculating the CV (coefficient of variation) of the relative peak area in all QC samples, and deleting the compounds with a CV greater than 30%. In the univariate analysis, fold change (FC) was obtained through fold change analysis, the *p*-value was determined by the *t*-test, and the *p*-value was corrected for false discovery rate (FDR) to obtain the *q*-value. Differential metabolites between groups were screened using the variable importance in the projection (VIP) values, FC, and *q*-values (VIP ≥ 1, FC ≥ 1.2 or ≤ 0.83, *q* < 0.05). Pathway enrichment analysis of differential metabolites was performed, and the significantly enriched pathways were determined with a *p*-value < 0.05.

### 2.6. The Correlation Analysis between Gut Microbiota and Differential Metabolites

The correlations between gut microbiota biomarkers and significantly differential metabolites were evaluated by Pearson correlation analysis and visualized in a heatmap using the package “gplots” in R. If *p*-value < 0.05, the correlation was considered to be significant. Furthermore, we used the Mantel test to evaluate the relationship between the key differential gut microbes and differential metabolites. The Mantel test was performed using the “vegan” package.

### 2.7. Statistical Analysis

IBM SPSS Statistics 25 software (Chicago, IL, USA) was used for one-way analysis of variance (ANOVA) and followed by Duncan’s multiple comparisons to obtain the growth performance (including body weight, body length, weight gain rate, and specific growth rate) of adult GFPs in three groups. The data are presented as means ± standard deviations (S.D.). A *p* < 0.05 indicated a significant difference, and *p* < 0.001 indicated a highly significant difference. The parameters were calculated as follows:

Weight gain (WG, %) = (mean final body weight-mean initial body weight)/mean initial body weight × 100

Specific growth rate (SGR, %/day) = [(Ln (final body weight) − Ln (initial body weight)/Culture period in days] × 100

## 3. Results

### 3.1. Growth Performance of GFP Families

After 92-day rearing, the 90 GFP families were collected for growth trait measurement (Figure 1). The three groups with different growth performance, including the high growth performance group with an average body weight of 41.03 ± 1.44 g, WG of 1616.82 ± 60.08%, and SGR of 3.09 ± 0.04 %/d, the middle growth performance group with an average weight of 37.46 ± 0.73 g, WG of 1467.45 ± 30.54%, and SGR of 2.99 ± 0.02 %/d, and the low growth performance group with an average weight of 33.34 ± 1.97 g, WG of 1294.82 ± 82.31%, and SGR of 2.86 ± 0.07 %/d, were selected as representatives (Table 1). The GFPs in Group H had significantly higher (*p* < 0.001) FBW, FBL, WG, and SGR than those in M and L groups. Moreover, the FBW, FBL, WG, and SGR of the GFPs in Group L were significantly lower (*p* < 0.001) than in Group M (Figure 2).

### 3.2. Gut Microbiota of GFP Families

#### 3.2.1. Composition of the Gut Microbiota within the Three Levels of Growth Performance

According to Chao1, ACE, observed species, and PD whole tree indexes, Group H had significantly higher richness, OTUs numbers, and characterization evenness of gut microbiota than Group L. However, the Shannon and Simpson indexes had slight differences among the three groups but they were not significant (Appendix A). This indicated that the overall diversity of microbiota in the three groups was similar. The distribution of OTUs in different groups showed that Group H had the most OTUs (1800), followed by Group M (1461) and Group L (796). All groups contained 378 core OTUs, and Group L had the highest ratio of core OTUs to unique OTUs (378/199) (Figure 3A). The NMDS diagram revealed obvious separations among the three groups (Figure 3B). In addition, PCA and PCoA analysis showed significant differences in the bacterial community among the three groups (Figure 3C,D).

#### 3.2.2. Identification of Differential Gut Microbiota within the Three Levels of Growth Performance

The histogram of LDA distribution and the branching diagram are shown in Figure 4. In Group H, the Firmicutes, Desulfobacterota, and Acidobacteriota at the phylum level were significantly enriched. At the class level, Bacilli, Clostridium, and Desulfovibrionia were enriched. At the order level, Bacteroidales, Lachnospirales, Peptostreptococcales, Tissierellales, and Desulfovibrionales were enriched. There were four main biomarkers at family level in Group H, including Lactobacillaceae, Lachnospiraceae, Peptostreptococcaceae, and Desulfovibrionaceae. At the genus level, *Lactobacillus*, *Romboutsia*, *Desulfovibrio*, *Blautia*, and *Lachnoclostridium* were detected. The biomarkers at the species level mainly included *Lactobacillus iners*, *Romboutsia ilealis*, and *Pseudomonas azotoformans*. Likewise, we found that the families Weeksellaceae, Flavobacteriaceae, and Vermiphilaceae were mostly enriched in Group L, and the genera *Gemmobacter*, *Cloacibacterium*, and *Flavobacterium* were also detected.

To further evaluate the differential gut flora among the three groups, the relative abundances of intestinal microbes in each group at the phylum and genus level are shown in Appendix A, respectively. The dominant phyla in all groups were Proteobacteria, of which the relative abundances in H, M, and L groups were 83.7%, 88.3%, and 92.8%, respectively, and Firmicutes, of which the relative abundances in H, M, and L groups were 14.2%, 9.7%, and 4.4%, respectively. The Firmicutes was more abundant in Group H than in other groups, while Proteobacteria was less abundant. A genus-level comparison of intestinal microbes revealed that the relative abundances of *Enterobacter*, *Lactobacillus*, *Escherichia-Shigella*, *Desulfovibrio*, *Romboutsia*, *[Ruminococcus] torques group* in Group H were significantly higher than in the other two groups, and those in Group M were higher than those in Group L. Notably, *Rikenellaceae_RC9_gut_group*, *Butyricicoccus,* and *CHKCI001* were only detected in Group H. *Faecalibacterium* (0.057%) and *Romboutsia ilealis* (0.11%) were more abundant in Group H than in the other two groups (Table 2). The heatmaps of the abundances of intestinal microbes among different groups further revealed the difference in the composition of intestinal microbes (Appendix A).

### 3.3. Metabolomics Analyses of GFP Families

#### 3.3.1. Identification of Differential Intestinal Metabolites among Three Groups

PCA showed that the samples from three groups were divided into three clusters and the samples from same group were clustered together according to the first two components in ESI^+^ and ESI^−^ scan modes (Figure 5A,B). The differential metabolites among the three groups are shown in Table 3. A total of 1173 metabolites with significant differences were identified among different groups in ESI^+^ mode. Meanwhile, 428 metabolites with significant differences were identified among different groups in ESI^−^ mode. Venn diagrams (Figure 5C,D) in positive and negative scan modes showed overlapping and specific differential metabolites identified from the pairwise comparisons, and 19 overlapping metabolites were detected among the three pairwise comparisons. Based on significantly different metabolites, the clustering analysis indicated obvious differences between H and L groups (Appendix A).

Compared to Group L, a total of 169 and 256 metabolites were upregulated, and 169 and 146 metabolites were downregulated in Group H and Group M, respectively. Compared to Group M, a total of 191 metabolites were upregulated, and 242 metabolites were downregulated in Group H. To screen out the key metabolites that probably mediated the growth performance of GFPs, we classified the functions of metabolites, and focused on 20 key differential metabolites (Table 4). Compared to M and L groups, the levels of spermidine, 4-oxoproline, adenosine, N-acetyl-l-phenylalanine, tryptophol, 8(s)-hydroxy-(5z,9e,11z,14z)-eicosatetraenoic acid, 13(s)-hotre, hydroxylysine, trimethylamine n-oxide, creatinine, and N-[(5s)-5-amino-5-carboxypentanoyl] cysteinyl-d-valine were significantly higher in group H (*p* < 0.05). Conversely, the levels of L-citrulline, P-dmea, maleamic acid, cytosine, 4-guanidinobutyric acid, 5-guanidino-2-oxopentanoic acid, 3-methoxytyramine, (9cis)-retinal, and 4-hydroxy-3-octaprenylbenzoic acid in Group L were much higher than in H and M groups.

#### 3.3.2. Key Roles of the Differential Metabolites in Growth Performance

The KEGG enrichment analysis revealed that ”D-arginine and D-ornithine metabolism”, “arginine biosynthesis”, “arginine and proline metabolism”, “tryptophan metabolism”, “amino acid biosynthesis”, ”ABC transport synthesis”, “TCA cycle”, “taurine and hypotaurine metabolism”, “alanine, aspartate, and glutamic acid metabolism”, “phenylalanine, tyrosine, and tryptophan biosynthesis”, “linoleic acid and purine metabolic pathways”, “pantothenic acid and CoA biosynthesis”, and “thiamine metabolism pathway” were significantly enriched (Appendix A). Likewise, we performed the enrichment analysis with 20 key metabolites, and they were enriched in the pathways, including “arginine and proline metabolism”, “PPAR signaling pathway”, “arachidonic acid metabolism”, and “retinol metabolism” (Figure 6). Compared to individuals from M and L groups, the upregulated metabolites in Group H were involved in “arginine and proline metabolism”, “arachidonic acid metabolism”, “alpha-Linolenic acid metabolism”, and “purine metabolism” etc. Likewise, the upregulated metabolites in Group L were enriched in “arginine biosynthesis” and “nicotinate and nicotinamide metabolism”. Interestingly, we found that the pathways associated with metabolism of amino acids and fatty acids were extensively detected based on the enrichment analyses. Hence, we inferred that the upregulated metabolites related to metabolism of amino acids and fatty acids may play critical roles in the growth of GFP.

### 3.4. Correlation Analysis between Gut Microbiota and Metabolites

To find the key microbiota and metabolites associated with growth, the correlation analysis of these two datasets was performed. The results indicated that there was a close correlation between the gut microbiota and metabolites in H and L groups (Figure 7A). The results of the Mantel test analysis were consistent with the correlation results (Figure 7B). The abundances of Clostridiaceae and *Lactobacillus* were significantly positively correlated with the levels of pyroglutamate, alpha-ketoglutaric acid, L-arginine, and D-ornithine. The abundance of *Bacteroides* was positively correlated with the levels of adenosine and taurine. Notably, the abundance of *Blautia* was highly positive correlated with the levels of adenosine, taurine, leucine, pyroglutamate, alpha-ketoglutaric acid, L-arginine, nicotinic acid, D-ornithine, and spermidine. In addition, it was negatively correlated with the levels of L-glutamic acid. However, among the main metabolites, alpha-ketoglutaric acid was significantly correlated with several gut microbiota, including Clostridiaceae, *Lactobacillus*, *Bacteroides*, *Blautia*, *Vibrio*, *CHKCI001* sp., *Bacillus,* and *Escherichia-Shigella*. Meanwhile, metabolite-to-metabolite correlation analysis revealed that the abundances of key differential metabolites, except L-glutamic acid, that were positively and significantly correlated.

### 3.5. The Roles of the Metabolites Related to Metabolism of Key Amino Acids and Fatty Acids in Growth Performance of GFP

The metabolite pathway enrichment and integrative analysis showed that the gut metabolites and their association with gut microbiota affected the growth performance of GFP by mediating the pathways of metabolism of amino acids and fatty acids (Figure 8). In “Purine metabolism”, adenosine, a key regulatory in energy metabolism, was significantly upregulated in Group H. Likewise, lots of metabolites, such as L-arginine, spermidine, creatinine, D-oxoproline, and D-ornithine, which mediate the downstream pathways, including arginine biosynthesis and metabolism, as well as taurine and hypotaurine metabolism, were upregulated in Group H. Importantly, all of arginine, taurine, and hypotaurine are associated with the biosynthesis of growth hormone in animals. Moreover, the unsaturated fatty acids (i.e., α-Linolenic acid and 13(s)-hotre) related to fatty acid metabolism were also upregulated in the fast-growing individuals. The up-regulated key metabolites that participate in arginine, taurine, and hypotaurine biosynthesis and metabolism, as well as metabolism of unsaturated fatty acids, play a vital role in the growth of GFP. They could be used as potential biomarkers for selecting the fast-growing prawns.

## 4. Discussion

### 4.1. Gut Microbiota Promotes the Growth and Metabolism of GFPs

The intestinal microbiota structure is influenced by the host genetics, and the host genetic effects on the gut microbiota are almost ubiquitous [34,35]. GFP growth rate was substantially increased through artificial selection and breeding, and its gut microbiota may also have been altered [36]. This study was the first to investigate the diversity and richness of intestinal bacterial communities with 16S rRNA sequencing. In this study, the differences in gut microbiota were compared with the GFP families in different growth performances. With the results of 16S rRNA sequencing, we found that the Firmicutes-to-Bacteroidetes ratio was higher in Group H than in Group M or L. Previously, Liu et al. [37] reported that Proteobacteria, Firmicutes, and Bacteroidetes were the dominant phyla in GFPs, which was consistent with our results. Furthermore, Turnbaugh et al. [38] found that Firmicutes promoted energy absorption by improving lipid metabolism. Bacteroidetes could increase carbohydrate metabolism to promote energy metabolism [39]. Previous studies [40,41] reported that Firmicutes could improve energy utilization in diets, and the Firmicutes-to-Bacteroidetes ratio was positively correlated with body weight gain in chickens. In crustaceans, a study also demonstrated that the abundance of Firmicutes in normal-growing shrimp was significantly higher than that in slow-growing shrimp (*p* < 0.01) [42]. Hence, we inferred that the high abundance of Firmicutes in Group H could help GFPs to absorb and store more energy, improving growth performance.

Additionally, we found that Lachnospiraceae was highly enriched in Group H, and the relative abundances of *Blautia* genus and *[Ruminococcus] Torques group* belonging to Lachnospiraceae were also the highest in Group H among the three groups. Lachnospiraceae produces butyric acid, which promotes the intestinal and general health of animals [43]. It has been reported that butyric acid could improve the growth performance of *L. vannamei* [44]. Notably, the relative abundance of *Faecalibacterium* was the highest in Group H. *Butyricicoccus* was only detected in Group H. Previous studies showed that *Butyricicoccus* and *Faecalibacterium* could produce butyrate from acetic acid, which promoted proliferation and differentiation of epithelium of the gut by releasing butyrate close to the epithelium [45]. These probiotics detected from Group H could account for the good growth performance of GFP. Meanwhile, Group H had the highest abundance of lactic acid bacteria species, such as *Lactobacillus*, which could regulate the intestinal environment, mucosal immunity, and maintain intestinal function [46,47]. The intestinal tracts of GFPs from the high growth performance group had the highest relative abundances of *Butyricicoccus*, *Faecalibacterium*, *Vibrio*, *Bacteroides*, and *Lactobacillus*, which could improve the intestinal digestion and also maintain gut health of the host.

### 4.2. Key Metabolites Play Important Roles in the Growth of GFPs

Intestinal microbiota-derived metabolites have been proposed to play key roles in nutrient digestion and absorption of aquatic animals [18]. Genetic variations could affect the intestinal metabolism, which in turn regulates the expression of genes related to metabolic processes [48]. Meanwhile, numerous physiological functions of the host are affected by the microbiota through its metabolic products. In this study, the abundance of spermidine, creatinine, adenosine, 8(s)-hydroxy-(5z,9e,11z,14z)-eicosatetraenoic acid, and 13(s)-hotre in Group H were significantly higher compared to other groups. Meanwhile, the levels of these five metabolites in Group L were lower than those in Group M. The abundance of L-citrulline in Group L was upregulated compared to Group H and M. These metabolites were involved in the metabolism of several key amino acids (i.e., arginine, taurine, and hypotaurine), fatty acid metabolism, and purine metabolism.

Spermidine and creatinine are involved in “arginine and proline metabolism”. Arginine is the precursor of the biosynthesis of spermidine [49], and could boost the level of glutathione (GSH) [50]. Moreover, spermidine was positively correlated with glutathione metabolism [51], which benefited intestinal integrity and improved its immune function [52]. Reduced spermidine levels lead to decreased glutathione levels [51]. Spermine and spermidine promoted intestinal proliferation and maturation, and they played a key role in development [53]. Except for the endogenous production, spermidine could be biosynthesized by gut bacteria [54]. Liu et al. [55] reported that higher creatinine levels were associated with growth. Creatinine was produced by the metabolism of creatine and related to “arginine and proline metabolism”. Proline and arginine could be produced by glutamine metabolism. However, glutamine could indirectly contribute to creatinine synthesis and nutrient absorption [56].

L-citrulline is associated with “arginine biosynthesis”. Citrulline is the sole precursor for arginine synthesis [57]. Biosynthesis of arginine could decrease L-citrulline levels by converting citrulline into arginine. Adenosine was enriched in “purine metabolism” in this study. It was widely accepted that adenosine was an energy metabolite [58]. Adenosine modulated a wide range of processes in many gastrointestinal cells [59], and was essential for ATP biosynthesis and CoA biosynthesis required for fatty acid metabolism [60]. Meanwhile, we found that 8(s)-hydroxy-(5z,9e,11z,14z)-eicosatetraenoic acid and 13(s)-hotre, which participated in the “arachidonic acid metabolism” and “alpha-linolenic acid metabolism”, were upregulated in Group H. “Arachidonic acid metabolism” and “alpha-linolenic acid metabolism” are related to fatty acid metabolism [61]. Furthermore, Uengwetwanit et al. [23] reported that shrimp growth and development were largely dependent on lipid metabolism. Meanwhile, the “PPAR signaling pathway” was also important in lipid metabolism and could regulate energy balance [62]. Therefore, we speculated that the metabolites related to “arginine and proline metabolism” and lipid metabolism may play important roles in the growth performances of GFPs.

Previously, Lin et al. [63] identified a number of important metabolites that were produced by gut microbiota such as amino acids. In this study, among the metabolites in the intestine of prawns between the H and L groups, the amino acids were the primary metabolic products of intestinal flora, including spermidine, D-ornithine, nicotinic acid, L-arginine, alpha-ketoglutaric acid, pyroglutamate, leucine, taurine, and adenosine, which played a key role in growth and development of GFPs. Notably, spermine and spermidine are formed from ornithine [64], and could also promote intestinal flora growth [65]. Alpha-ketoglutaric acid (AKG) was a TCA cycle intermediate. The TCA cycle is a metabolic nexus that links the metabolism of carbohydrates, fats, and proteins [66]. Meanwhile, it is an important metabolic pathway for bacteria, and bacteria are reliant on the TCA cycle for energy production [67]. It has been demonstrated that taurine could improve the utilization rate of protein in feed [68], increase protein deposition in aquatic animals [69], and promote growth in animals [70]. Taurine affects the activities of various digestive enzymes of fish, promotes metabolism, increases survival rate, increases growth rate, and helps fish develop morphologically [71]. Overall, Group H possessed more superior metabolites and had better digestion, absorption, and metabolism than Group L.

Animal hosts could absorb nutrients via the fatty acid metabolisms with Firmicutes [72]. Several members of *Lactobacillus* mediated polyamine biosynthesis through the biosynthesis and transport pathway of spermine [73]. Spermine is involved in arginine and proline metabolism. In this study, compared to Group L, spermine was upregulated in Group H, and positively promoted the growth of GFPs. Additionally, Seenivasan et al. [74] found that probiotics could promote the growth and energy budget performance of *M. rosenbergii* post larvae. Liu et al. [75] indicated that a majority of the properties of *Blautia* were related to its potential probiotic functions. Meanwhile, previous findings suggested that intestinal bacteria, including *Blautia*, were important in maintaining health by providing ornithine [76]. Furthermore, in this study, the functions of other metabolites were significantly correlated with those of *Blautia*, such as pyroglutamate and leucine, which need to be investigated further. Intestinal microbiota and metabolites interacted to affect the growth performance of GFPs.

## 5. Conclusions

This was the first study to use 16S rRNA sequencing and metabolomic methods to study intestinal bacteria composition, metabolite characteristics, and their interactions in GFP families with different growth performances. The results showed that some critical intestinal bacteria, including *Lactobacillus*, *Blautia,* etc., were positively correlated with D-ornithine, L-arginine, alpha-ketoglutaric acid, and pyroglutamate. In particular, the metabolites related to arginine, taurine, and hypotaurine biosynthesis and metabolism, as well as metabolism of unsaturated fatty acids, play vital roles in the growth of GFP. These were significantly higher in the fast-growing prawns. Importantly, these intestinal microbiota and metabolites could positively affect intestinal digestion and absorption, and then promote GFP growth. Overall, this study provides a novel insight into the growth regulation of GFP families, and presents the feasibility of the development of probiotic feeds.

## Figures and Tables

**Figure 1 animals-13-01539-f001:**
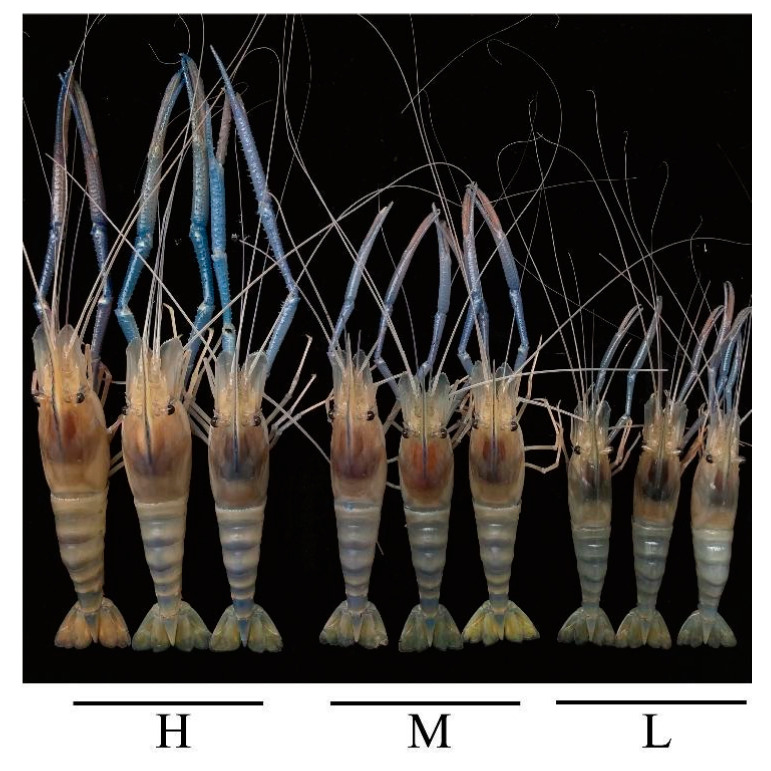
Morphological characteristics of the three growth levels of GFP. H, high growth performance level; M, middle growth performance level; L, low growth performance level.

**Figure 2 animals-13-01539-f002:**
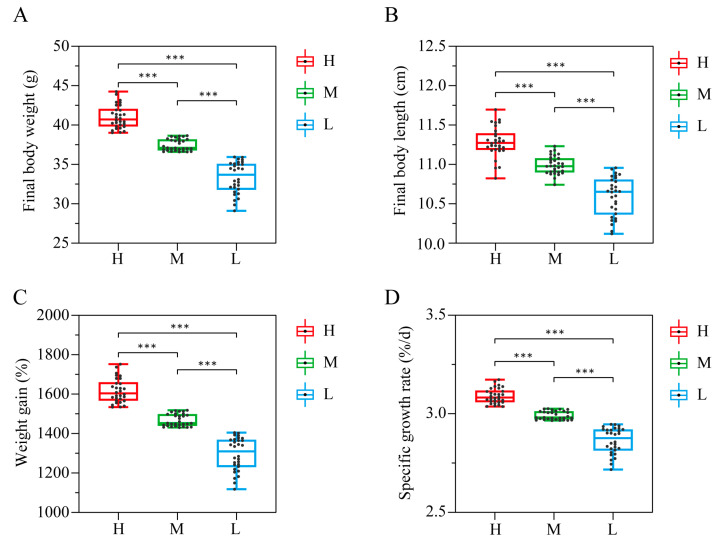
Box plots of growth performances in GFP families (n = 30). Note: (**A**) FBW, final body weight. (**B**) FBL, final body length. (**C**) WG, weight gain. (**D**) SGR, specific growth rate. *** indicates extremely significant differences (*p* < 0.001). H, high growth performance level; M, medium growth performance level; L, low growth performance level.

**Figure 3 animals-13-01539-f003:**
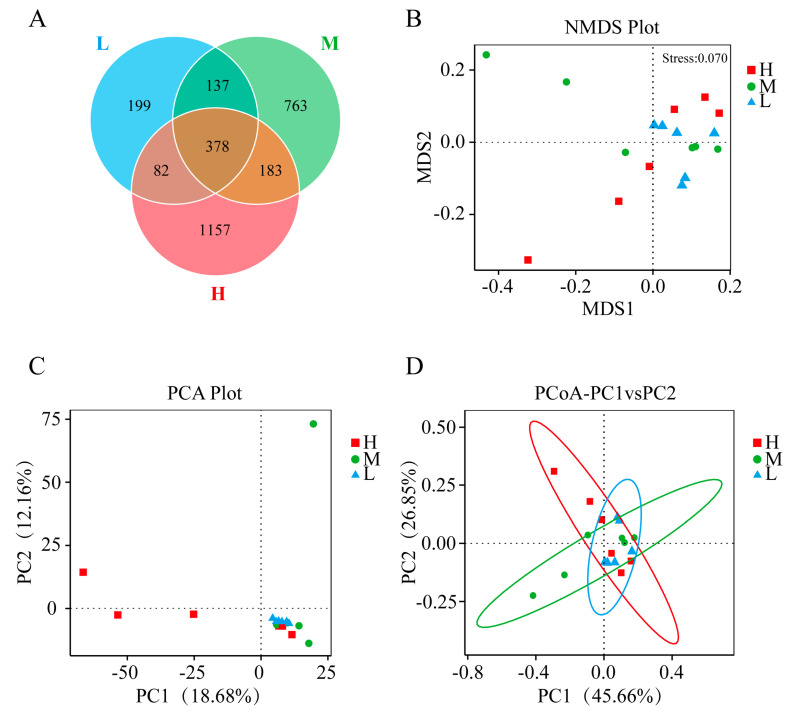
(**A**) Venn diagram of OTUs in GFP families with three growth performances; (**B**) based on Bray–Curtis distances, nonmetric multidimensional scaling (NMDS) analysis in GFP families with three growth performances; (**C**) principal component analysis (PCA) in GFP families with three growth performances; (**D**) principal coordinates analysis (PCoA) in GFP families with three growth performances. H, high growth performance level; M, medium growth performance level; L, low growth performance level.

**Figure 4 animals-13-01539-f004:**
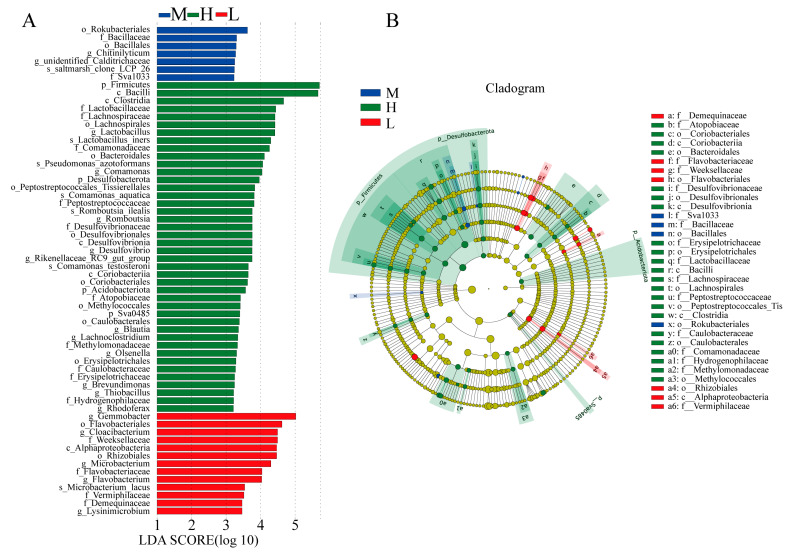
(**A**) A histogram of the distribution of linear discriminant analysis (LDA) values among GFP families with three growth performance groups; (**B**) an evolutionary branching diagram of species abundance among GFP families with three growth performance groups. LDA score > 2.2. In a branching diagram, node size represents the average relative abundance of the taxon. The yellow nodes represent taxa that are not significantly different among the three groups, whereas the green, red, and blue nodes indicate that these taxa are significantly different. And this color represents a higher abundance in the corresponding sample group. Identifies taxa with significant differences using letters. H, high growth performance level; M, medium growth performance level; L, low growth performance level.

**Figure 5 animals-13-01539-f005:**
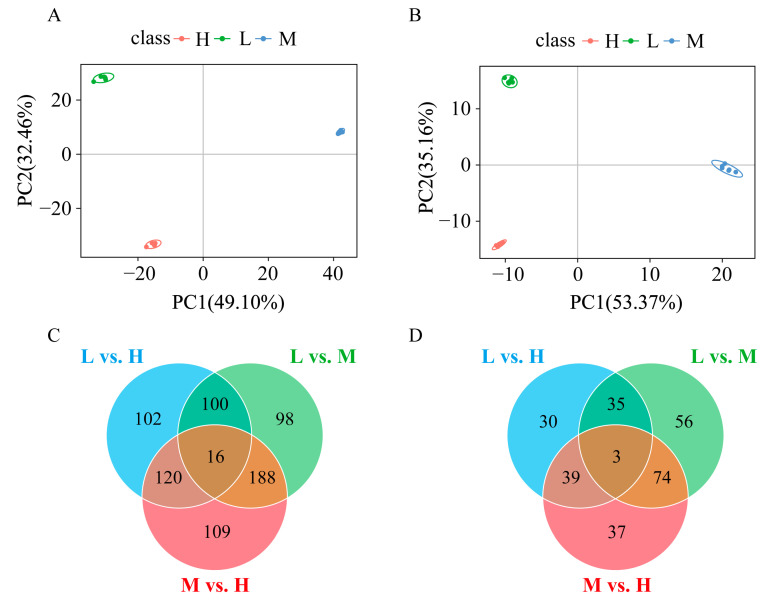
The graphs of PCA scores in ESI^+^ and ESI^−^ scan modes for three growth performance groups in (**A**,**B**). The Venn diagrams in ESI^+^ (**C**) and ESI^−^ (**D**) scan modes show the number of common and specific metabolites that differ significantly among the three pairwise comparisons groups. The three pairwise comparisons included L vs. H, L vs. M, and M vs. H. H, high growth performance level; M, medium growth performance level; L, low growth performance level.

**Figure 6 animals-13-01539-f006:**
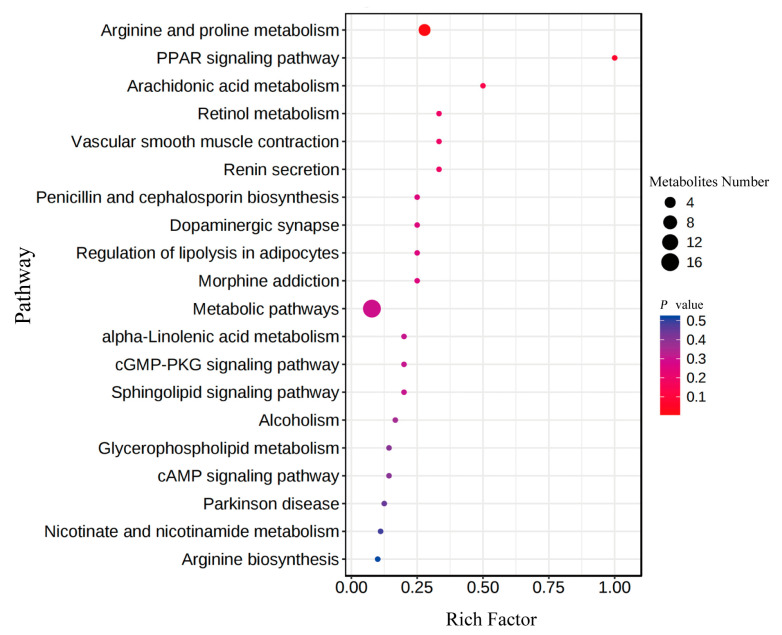
KEGG pathway enrichment analysis of the differential metabolites identified among three growth performance groups of GFPs.

**Figure 7 animals-13-01539-f007:**
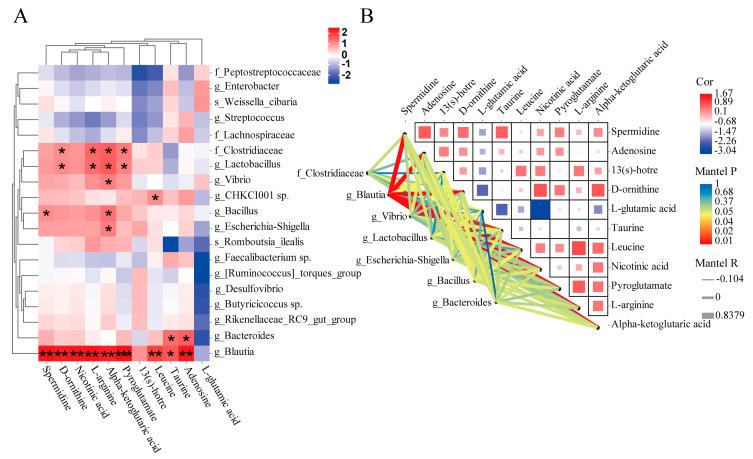
(**A**) Heatmap of correlation coefficients between the key gut microbiota and metabolites identified between H and L groups. * Indicates significantly positively correlated; ** indicates highly significantly positively correlated; *** indicates the highest significantly positively correlated; (**B**) correlation between differential metabolites and correlation between the key differential gut microbiota and metabolites in H and L groups using Mantel test. The color patch in each square of the heatmap represents the positive or negative correlation coefficient between metabolites, and the patch size represents the absolute value of the correlation coefficient. The key intestinal microbiota is associated with differential intestinal metabolites one by one. The Line thickness indicates the strength of the correlation, and the line color indicates the degree of significance. H, high growth performance level; L, low growth performance level.

**Figure 8 animals-13-01539-f008:**
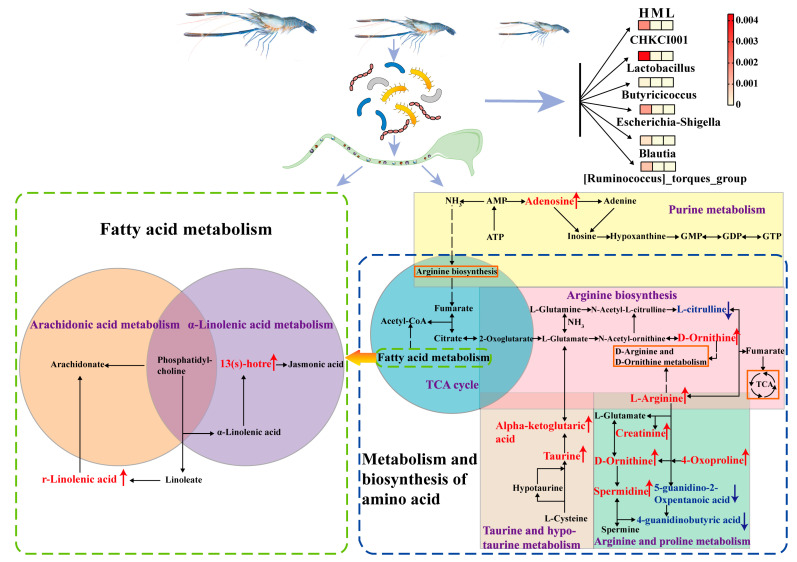
Mapping of key differential metabolites between H and L groups in the relevant KEGG pathways. The “Arachidonic acid metabolism pathway” is shown in khaki, the “α-Linolenic acid metabolism pathway” is in purple, the “Arginine and proline metabolism pathway” is in green, the “Arginine biosynthesis pathway” is in pink, the “Purine metabolism pathway” is in yellow, the “Taurine and hypotaurine metabolism pathway” is in gray, and the “TCA cycle pathway” is in blue. The up- and down-regulated metabolites are presented by red and blue arrows, respectively. And their font colors are also presented by red and blue, respectively. Dashed arrows indicate the connectivity to another pathway. The color of each square next to the bacteria represents the intensity of the bacteria abundance, the red indicates the highest abundance, and the yellow indicates the lowest abundance.

**Table 1 animals-13-01539-t001:** Growth performance of *M. rosenbergii* families under the same feeding conditions for 92 days.

Sample Name	FBW (g)	FBL (cm)	WG (%)	SGR (%/d)
Group H	41.03 ± 1.44 ^A^	11.28 ± 0.19 ^A^	1616.82 ± 60.08 ^A^	3.09 ± 0.04 ^A^
Group M	37.46 ± 0.73 ^B^	10.99 ± 0.11 ^B^	1467.45 ± 30.54 ^B^	2.99 ± 0.02 ^B^
Group L	33.34 ± 1.97 ^C^	10.59 ± 0.24 ^C^	1294.82 ± 82.31 ^C^	2.86 ± 0.07 ^C^

Note: FBW, final body weight; FBL, final body length; WG, weight gain; SGR, specific growth rate. ^A,B,C^ Different letters indicate extremely significant differences (*p* < 0.001). H, high growth performance level; M, middle growth performance level; L, low growth performance level.

**Table 2 animals-13-01539-t002:** Relative abundances of differential bacterial communities in GFP families with three growth performance groups.

Phylum	Family	Genus	Species	Group H		Group M		Group L	
				Mean (%)	S.D. (%)	Mean (%)	S.D. (%)	Mean (%)	S.D. (%)
Proteobacteria				83.66	8.242	88.29	6.424	92.80	2.199
Firmicutes				14.20	8.656	9.665	5.391	4.370	2.558
Bacteroidota				0.397	0.335	0.408	0.506	0.797	0.933
Desulfobacterota				0.204	0.221	0.012	0.009	0.006	0.005
Firmicutes	Lactobacillaceae			0.470	0.705	0.021	0.010	0.006	0.005
Firmicutes	Lachnospiraceae			0.502	0.333	0.027	0.025	0.007	0.007
Firmicutes	Peptostreptococcaceae			0.136	0.111	0.036	0.036	0.010	0.004
Firmicutes	Lactobacillaceae	*Lactobacillus*		0.431	0.700	0.007	0.005	0.001	0.001
Firmicutes	Lachnospiraceae	*CHKCI001*		0.188	0.221	——	——	——	——
Firmicutes	Peptostreptococcaceae	*Romboutsia*		0.111	0.114	0.003	0.003	0.001	0.001
Firmicutes	Lachnospiraceae	*[Ruminococcus]_torques_group*		0.095	0.102	0.0002	0.001	——	——
Firmicutes	Lachnospiraceae	*Blautia*		0.043	0.022	0.003	0.005	0.0005	0.001
Firmicutes	Ruminococcaceae	*Faecalibacterium*		0.057	0.057	0.007	0.007	0.005	0.005
Firmicutes	Butyricicoccaceae	*Butyricicoccus*		0.014	0.016	——	——	——	——
Firmicutes	Streptococcaceae	*Streptococcus*		0.017	0.010	0.010	0.008	0.004	0.004
Proteobacteria	Enterobacteriaceae	*Enterobacter*		2.045	2.123	1.426	0.327	1.356	0.554
Proteobacteria	Enterobacteriaceae	*Escherichia-Shigella*		0.156	0.291	0.009	0.009	0.003	0.002
Proteobacteria	Vibrionaceae	*Vibrio*		0.018	0.017	0.014	0.010	0.005	0.004
Bacteroidota	Rikenellaceae	*Rikenellaceae_RC9_gut_group*		0.129	0.246	——	——	——	——
Bacteroidota	Bacteroidaceae	*Bacteroides*		0.037	0.041	0.002	0.002	0.001	0.001
Desulfobacterota	Desulfovibrionaceae	*Desulfovibrio*		0.126	0.159	0.002	0.002	0.001	0.002
Firmicutes	Peptostreptococcaceae	*Romboutsia*	*Romboutsia_ilealis*	0.111	0.114	0.003	0.003	0.001	0.001

**Table 3 animals-13-01539-t003:** Summary of the numbers for differential metabolites in ESI^+^ and ESI^−^ scan modes among three comparison groups.

Mode	Pairwise Comparison	Total Differential Metabolites Number	Up/Downregulated Differential Metabolites Number	Identification Level 1 Differential Metabolites Number		Identification Level 2 Differential Metabolites Number	
				Upregulated	Downregulated	Upregulated	Downregulated
ESI^+^	Group L vs. Group H	338	169/169	3	8	9	10
	Group L vs. group M	402	256/146	6	8	23	13
	Group M vs. Group H	433	191/242	5	8	9	22
ESI^−^	Group L vs. Group H	107	60/47	1	0	4	5
	Group L vs. group M	168	87/81	2	0	5	0
	Group M vs. Group H	153	94/59	9	1	13	6

Note: H, high growth performance level; M, medium growth performance level; L, low growth performance level.

**Table 4 animals-13-01539-t004:** The abundance of key intestinal metabolites among three growth performance groups (VIP ≥ 1; FC ≥ 1.2 or FC ≤ 0.83; *p* < 0.05).

Name	Pathway	Group H	Group M	Group L
		Mean	Mean	Mean
Spermidine	Arginine and proline metabolism	7,568,085.52	503,668.63	297,921.56
L-citrulline	Arginine biosynthesis	12,351,334.96	28,501,478.19	50,349,665.38
4-oxoproline	Arginine and proline metabolism	34,504,224.16	21,513,189.36	14,195,014.97
Adenosine	Purine metabolism	20,948,251.26	16,955,727.72	5,034,361.71
N-acetyl-l-phenylalanine	Phenylalanine metabolism	16,939,441.79	9,720,963.52	8,590,198.80
Tryptophol	Tryptophan metabolism	3,882,503.94	2,171,436.62	1,993,910.23
8(s)-hydroxy-(5z,9e,11z,14z)-eicosatetraenoic acid	Arachidonic acid metabolism	12,940,493.38	12,410,918.29	8,284,469.84
13(s)-hotre	alpha-Linolenic acid metabolism	9,965,759.94	5,687,309.98	5,470,815.92
Hydroxylysine	Lysine degradation	1,790,922.13	1,040,770.98	576,601.17
P-dmea	Glycerophospholipid metabolism	2,420,117.054	5,503,074.12	6,928,635.32
Maleamic acid	Nicotinate and nicotinamide metabolism	3,819,384.43	7,534,743.65	9,223,492.68
Trimethylamine n-oxide	Metabolic pathways	11,407,950.24	9,534,357.45	2,708,587.02
Cytosine	Pyrimidine metabolism	26,885,730.18	32,999,513.02	67,604,815.23
Creatinine	Arginine and proline metabolism	24,423,233.36	18,256,455.97	15,024,899.59
4-guanidinobutyric acid	Arginine and proline metabolism	4,072,488.34	4,784,183.04	7,288,469.76
5-guanidino-2-oxopentanoic acid	Arginine and proline metabolism	5,655,838.31	8,563,793.46	10,158,770.44
N-[(5s)-5-amino-5-carboxypentanoyl] cysteinyl-d-valine	Metabolic pathways	2,077,361.39	2,046,182.39	1,352,700.594
3-methoxytyramine	Tyrosine metabolism	622,589.31	1,369,607.54	1,611,085.97
(9cis)-retinal	Retinol metabolism	13,138,741.23	20,809,948.49	30,180,669.50
4-hydroxy-3-octaprenylbenzoic acid	Ubiquinone and another terpenoid-quinone biosynthesis	7,617,111.30	20,111,478.28	20,269,682.87

Note: H, high growth performance level; M, medium growth performance level; L, low growth performance level.

## Data Availability

All supporting data are included within the main article and Appendix A.

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
