# Peer review of "Alterations of the Gut Microbiota and Metabolomics Associated with the Different Growth Performances of Macrobrachium rosenbergii Families"

_animals, 2023, doi:10.3390/ani13091539_

Round 1

Reviewer 1 Report

This manuscript investigates the correlation between growth rates, gut microbiome compositions and gut metabolite compositions in Giant Freshwater prawns. The manuscripts finds some interesting links between these parameters, which may be useful for breeding and feed manufacturing purposes. Overall, I find that the manuscript is interesting, well written and provides solid evidence for the conclusions drawn. Somehow I feel that the decision to lump the prawn families into three groups based on their growth performance leads to a loss of information compared to making correlations based on the whole range of growth rates. However, there may be some statistical reasons for doing this that I am not aware of. In general, the figure legends are very brief, making it difficult to interpret the figures. Some more explanatory text would be useful.

Line 15 + 28: without having read the paper yet, it is impossible to know what “etc.” means.

Line 61: Does not read well.

Line 111: What is meant by “should be”? It must be known if water quality was monitored daily?

Line 134: “A total of” should probably be deleted?

Line 157: Is “splicing” the right word? I would suggest “merging” instead.

Figure 2: Since the categorization into H, M and L is based on the growth rate, it is not surprising that there is a significant difference in growth parameters across these three groups. I think this figure is pointless unless I have misunderstood something.

Table 2: Legend does not match the table content.

Line 375: “Positively” instead of “positive”.

Figure 8: It is stated in the legend that “up- and down-regulated metabolites are presented by red and blue arrows, respectively”. It needs to be clarified what is meant by up- and down-regulated. Metabolites that are up-regulated in one group will be down-regulated in the group compared to, so these are relative rather than absolute terms.

Figure 8: It is stated in the legend that “the color of each square next to the bacteria represents the intensity of the bacteria abundance”. It is, however, not clear what the dimension is for the values given, and it is also not clear what the three boxes for each bacterium represents (the three groups, i.e. L, M and H?).

Figure S3: This figure could use some explanatory text, as it is unclear what it shows.

Line 531 - 532: Table S2 is mentioned twice – should the last one perhaps be S3?

Supplementary tables: I was not able to find a way to download the supplementary tables.   

Apart from the few places mentioned above, I did not detect any issues with the language.

Reviewer 2 Report

The present work explores the metabolomic and microbiome differences related to growth performance of the aquaculture prawn Macrobrachium rosenbergii . Overall, the present study is well written and its content is also clear. I only have a few concerns regarding the exaggerated number of tables and figures presented, I recommend that some go to the attachments. Indeed, some figures have a lot of information and may be disaggregated into several figures for better reading of the results, namely the figures 4 and 7. I only have a few more comments:

Line 134: Please clarify: A total of Before the sampling”, prawns from the 9 selected GFP families were starved.

Figure 2 – insert in the figure caption the meaning of H, M, L. The figure must be self-explanatory. Enlarge the letter size of y axis.

Figure 3. enlarge graph axis letter numbers…meaning of H. M L.

Reviewer 3 Report

It is a very interesting work.

Please define clearly the criteria to establish the 3 levels of growth performance. 

Reviewer 4 Report

The present manuscript investigated the key gut microbiota and metabolites associated with the growth performances of Macrobrachium rosenbergii families. Overall, this study contribute to figuring out the landscape of the gut microbiota and metabolites associated with growth performance of GFP families and provided an important basis for selective breeding of GFP growth traits. This is a well-designed, and clear manuscript. The figures were well displayed and enhanced the readability and comprehension of the results to a great extent. I recommended it to be accepted after minor revision.

1) The highlight of this experiment is that the correlation analysis between intestinal microbiota and metabolomics reveals the mechanism of the growth difference of Macrobrachium rosenbergii. However, in the introduction, authors only introduced the overview of intestinal microbiota and metabolomics, and did not summarize the research progress of the correlation analysis between these two technical analysis. Please add this part.

2) In 4.1, authors discussed the possible impact of changes in the abundance of intestinal flora on the digestion, absorption, and metabolism of prawns, but did not detect phenotypic indicators such as digestion and absorption-related enzyme activities. Therefore, the title of 4.1 is inappropriate and needs to be reconsidered by the authors.

3) Line 506-513, the discussion based on the results of the correlation analysis between intestinal microbiota and metabolomics is less and needs to be further improved.

4) Page 1 line 17, “prawn” should be “prawns”.

5) Page 3 line 131. Please transfer the phenotype picture of Figure 1 to result 3.1.

6) Page 4 line 140, “freezed” should be “froze”.

7) Page 15 line 450. Change “resulting in weight gain” to “improving growth performance”.

8) Page 17 lines 521-523. reference 68 should be moved to the above of reference 67.

9) Please correct the numbers on pages 16 and 17.

10) In the discussion, the author should add the statements how the metabolites and gut microbiota are determined by the genetic difference. It could explained that the growth difference of the GFP families were closely related to the metabolites and gut microbiota, which are also affected by genetics.

The English writing of MS should be improved.

Round 2

Reviewer 4 Report

The author has resolved my doubts and I have no further opinions.

The author has resolved my doubts and I have no further opinions.